# Characteristics of Turbulence in the Downstream Region of a Vegetation Patch

**Masoud Kazem [1], Hossein Afzalimehr [1,*] and Jueyi Sui [2]**

[1] Faculty of Civil Engineering, Iran University of Science and Technology, Tehran 16846-13114, Iran; masoud_kazem@cmps2.iust.ac.ir
[2] School of Engineering, University of Northern British Columbia, Prince George, BC V2N 4Z9, Canada; Jueyi.Sui@unbc.ca
* Correspondence: hafzali@iust.ac.ir; Tel.: +98-913-2175524

**Abstract:** In presence of vegetation patches in a channel bed, different flow–morphology interactions in the river will result. The investigation of the nature and intensity of these structures is a crucial part of the research works of river engineering. In this experimental study, the characteristics of turbulence in the non-developed region downstream of a vegetation patch suffering from a gradual fade have been investigated. The changes in turbulent structure were tracked in sequential patterns by reducing the patch size. The model vegetation was selected carefully to simulate the aquatic vegetation patches in natural rivers. Velocity profile, TKE (Turbulent Kinetic Energy), turbulent power spectra and quadrant analysis have been used to investigate the behavior and intensity of the turbulent structures. The results of the velocity profile and TKE indicate that there are three different flow layers in the region downstream of the vegetation patch, including the wake layer, mixing layer and shear layer. When the vegetation patch is wide enough ($D_v/D_c > 0.5$, termed as the patch width ratio, where $D_v$ is the width of a vegetation patch and $D_c$ is the width of the channel), highly intermittent anisotropic turbulent events appear in the mixing layer at the depth of $z/H_v = 0.7{\sim}1.1$ and distance of $x/H_v = 8{\sim}12$ (where $x$ is streamwise distance from the patch edge, z is vertical distance from channel bed and $H_v$ is the height of a vegetation patch). The results of quadrant analysis show that these structures are associated with the dominance of the outward interactions (Q1). Moreover, these structures accompany large coherent Reynolds shear stresses, anomalies in streamwise velocity, increases in the standard deviation of TKE and increases in intermittent Turbulent Kinetic Energy ($TKE_i$). The intensity and extents of these structures fade with the decrease in the size of a vegetation patch. On the other hand, as the size of the vegetation patch decreases, von Karman vortexes appear in the wake layer and form the dominant flow structures in the downstream region of a vegetation patch.

**Keywords:** coherent Reynolds shear stress; intermittent turbulence; mixing layer; quadrant analysis; vegetation patches

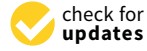

## 1. Introduction

Submerged vegetation patches are fundamental participants of aquatic ecosystems, providing important habitats for fauna, including fishes [1–3]. As the reduction in the aquatic vegetation is a global concern, researchers have conducted studies to investigate the concept of a large-scale environmental issue [4–6]. However, while most of the current hydraulic-orientated research is focused on the flow characteristics in the canopy region of fully vegetated channels, finite dimensional vegetation patches with no fully developed region downstream of vegetation patches are very common in natural rivers. These sparse vegetation patches have a substantial effect on flow structures depending on the dimensions of the vegetation patch [7–9]. The effect of vegetation patch dimensions on flow structures has significant consequences for the ecosystem through various mechanisms. For example, an intermediate coverage (10–40%) of vegetation may promote high species richness

due to the conditions provided by the optimal behavior of flow [10–12]. In addition, flow structures around a vegetation patch have a significant impact on the sediment transport, and consequently on the topography of the river bed [13–15]. Overall, the interaction between flow and vegetation in natural channels has attracted a lot of attention from hydraulic researchers. To mention only a few, the most recent research works have been conducted by researchers [16–18], in which various aspects of this interaction have been investigated.

Given the importance of the research topic, this present study aims to assess the effects of different sizes of vegetation patch on the flow characteristics in the downstream region of vegetation patches. The sizes of vegetation patches range from full coverage across the entire channel to a narrow vegetation patch located in the middle of the channel.

## 2. Flow Structures behind Vegetation Patches

Compared to a rough bed without a vegetation patch, the presence of a vegetation patch in a channel results in changes in the flow structures and momentum transfer. Therefore, flow resistance and turbulence characteristics are expected to differ in the downstream regions of vegetation patches [19]. To assess the characteristics of turbulent flow, the following methodologies are commonly used: spectral analysis of velocity time series, determination of Reynolds stresses, and Turbulent Kinetic Energy (TKE), as well as quadrant analysis [20–22]. The analysis of the coherent Reynolds shear stress is another method that is used to determine coherent turbulent structures. However, this approach has hardly been used in hydraulic-oriented works due to the difficulties in calculating the phase velocity.

In order to make clear the terminologies used in the above-mentioned approaches, an introduction is presented in this section alongside the literature review. In the classic terminology of turbulences, for a sufficiently long duration of velocimetry ($T$), and for the velocity component of $u_i(t)$, the average velocity ($\overline{u}_i$) is defined as:

$$\overline{u}_i = \lim_{T \to \infty} \frac{1}{T} \int_0^T u_i(t) dt \tag{1}$$

Considering $i = 1, 2, 3$ as the $x$, $y$ and $z$ directions in the Cartesian coordinate system, $\overline{u}$, $\overline{v}$ and $\overline{w}$ are defined as the averages of velocities in the $x$, $y$, and $z$ directions, respectively. Consequently, the velocity fluctuations are defined by:

$$\begin{aligned} u' &= u(t) - \overline{u} \\ v' &= v(t) - \overline{v} \\ w' &= w(t) - \overline{w} \end{aligned} \tag{2}$$

Then, the Turbulent Kinetic Energy (TKE) is also calculated as:

$$\text{TKE} = 0.5\left(\overline{u'^2} + \overline{v'^2} + \overline{w'^2}\right) \tag{3}$$

In general, as was reported by researchers, a sharp increase in TKE is linked to the presence of vortexes [23–26]. In partially covered channels with emergent vegetation patches, where the flow is almost 2D, the flow structure in the downstream region of vegetation patches is dominated by the horizontal von Karman vortexes developed by instabilities in the trailing edges of the vegetation patch [27,28]. The distance between the trailing edge of the barrier and the formation of von Karman vortexes is known as $L_{kv}$. According to reported research based on TKE analysis, for a relatively dense vegetation patch with a diameter of $D_v$, in which the leakage velocity from the dense vegetation patch is negligible, $L_{kv} \approx 2.5 D_v$ [27,28]. On the other hand, the 3D flow structures in the downstream region of a submerged vegetation patch are more complicated. In addition to the constant presence of a vertical recirculation zone beyond the vegetation patch, the presence and magnitude of the von Karman vortexes in the horizontal plane depends on

the geometry of the vegetation patch [28–30]. For instance, the presence of von Karman vortexes at a depth of $H_v/H > 0.55{\sim}0.7$, where $H_v$ is the height of the vegetation patch and $H$ is the flow depth [28,30]. On the other hand, results indicated that the dominant structure of a wide submerged path occurs in the vertical plane, as vortexes forming a vertical recirculation zone [28,29]. The distance between the trailing edge of the barrier and the forming of the vertical vortex is known as $L_{kv}$. For a solid submerged barrier, there is no gap between the trailing edge of the barrier and the forming vortex, and the length of wake behind the barrier is zero ($L_{vv} = 0$) [31]. However, for a porous vegetation patch, the velocity of flow passing through the vegetation patch may delay the formation of the recirculating region ($L_{vv} \neq 0$). The reported value of $L_{vv}$ ranges between $1H_v$ and $5H_v$ depending on the porosity and diameter of the vegetation patch [28,29]. As reported by Liu et al. (2018), the value tends to be about $1H_v$ in a relatively wide vegetation patch, and increases up to $5H_v$ in vegetation patches with a low blockage ratio [28]. These results indicate that $L_{vv}$ and $L_{kv}$ are affected by the vegetation patch dimensions of $h$ and $D$, respectively. In addition, the bleed velocity that penetrates into the wake region can also increase these values considerably [23,31].

In addition to the above-mentioned classic variables, coherent turbulent variables have become fundamental terms of flow analysis in recent years [32–34]. According to the triple decomposition approach, the instantaneous velocity can be written as [35]:

$$u(x,t) = \overline{u}(x) + \widetilde{u}_c(x,t) + u_r(x,t) \tag{4}$$

where $u(x,t)$ is the instantaneous velocity, $u_r$ is the deflection of velocity (known as $u'$ in the classic approach), and $\overline{u}(x)$ is the classic average of velocity during time period $T$. In addition to the classic averaging method, the periodic phase average can be defined as:

$$\langle u(x,t)\rangle = \lim_{N\to\infty} \frac{1}{T_P} \sum_1^N u_i(x, t + iT_P) \tag{5}$$

where $T_p$ is the period of occurrence of a coherent structure and is equal to $1/f_d$, where $f_d$ is the dominant frequency of the occurrence. Generally, when a coherent event occurs in a particular region of flow, $f_d$ will reach an obvious peak in the power density spectrum of the related time series in the same region. For a time series with a length of $T$, $N$ is the number of cycles with a period of $T_p$, which can be calculated as $N = T/T_p$. Following these definitions, the coherent velocity deflection is:

$$\widetilde{u}_c(x,t) = \langle u(x,t)\rangle - \overline{u}(x) \tag{6}$$

Considering the same approach for other components of velocity, the coherent Reynolds shear stress can be written as:

$$\tau_c = -\rho\langle \widetilde{u}_c \widetilde{w}_c\rangle \tag{7}$$

The total coherent and non-coherent Reynolds shear stress is defined as follows:

$$\tau_r = -\rho\overline{u_r}\overline{w_r} \tag{8}$$

Although it is very rare to implement $\tau_c$ in studies of river hydraulics with the presence of vegetation patches, a few research works have been conducted to study the variation in $\tau_r$ in river hydraulics in the presence of vegetation patches. For instance, in the downstream region of a submerged vegetation patch, there is a high gradient pattern of Reynolds shear stress in the vertical plane, which increases toward the surface of the flow [36]. This phenomenon is induced by the high-speed flow that passes above the submerged vegetation patch and causes a considerable velocity gradient when compared with the low velocity flow leaking through the vegetation patch. The region with a high velocity gradient is known as the shear layer. This region of flow has particular turbulence characteristics that will be studied in this paper.

As another method of analysis, the decomposition of bursting events (known as quadrant analysis in 2D space and octant analysis in 3D space) is also widely applied to determine the dominant turbulent events in the presence of both emerged and submerged vegetation patches [22,37–39]. The occurrence probability of event $k$ is calculated as the normalized occurrence frequency, $f_k$, for a particular class of events related to different classes of events [40]:

$$f_k = \frac{n_k}{N} \text{ with } N = \sum_{1}^{4} n_k \ k = 1, 2, 3, 4 \tag{9}$$

where $n_k$ is the number of events belonging to class $k$, and $N$ is the total number of events.

For the quadrant analysis, a "hole" region has been proposed in the majority of previous studies. In a "hole" region, the event must be filtered and not be considered. In octant analysis, this threshold can be written as:

$$\left| u'(t)w'(t) \right| > C_H \left| \overline{u'w'} \right| \tag{10}$$

For the threshold parameter $C_H$, the low-intensity events below a certain limit were omitted, which were scaled by the average of the velocity fluctuations. The high value of $C_H$ implies the selection of the strongest events, but the total number of instantaneous $[u'(t) \ w'(t)]$ decreases so much that the contribution region in each quadrant becomes meaningless. For quadrant analysis, the threshold level $C_H = 1$ is suggested for use to reach a good compromise between the clear identification of the events and the preservation of a number of instantaneous events of each class [40,41]. In this paper, $C_H$ is set as 1. The classes of events are also defined as:

➢ Q1—occurrence of the outward interaction when $u' > 0$ and $w' > 0$;
➢ Q2—occurrence of the ejection when $u' < 0$ and $w' > 0$;
➢ Q3—occurrence of the inward interaction when $u' < 0$ and $w' < 0$;
➢ Q4—occurrence of the sweep when $u' > 0$ and $w' < 0$.

To date, most of the reported research has applied quadrant analysis to study coherent flow structures within or above a vegetation patch. However, there are a few published studies in which quadrant analysis has been conducted to assess the coherent flow structures in the downstream region of a vegetation patch. Okamoto and Nezu (2013) point out that in the region immediately behind the submerged vegetation patch, there is a small margin of ejection dominancy around the top edge [7]. Similarly, based on data collected from experiments performed on dryland vegetation in a wind tunnel, Mayaud et al. (2016) revealed that there were elevated frequencies of Q2 (ejection) and Q4 (sweep) events in the immediate toe of the vegetation patch [42]. In contrast, it is reported the dominance of outward and inward interactions in the shear layer induced by the flow passing above the vegetation patch, which is a notable characteristic of the flow in the downstream region of a submerged vegetation patch in a more distant region [38]. It must be mentioned that the relation between quadrant occurrences and velocity structures is an interesting research topic that has received attention from researchers recently. Wang et al. (2019) established interconnections between the classic definition of vortex groups and quadrant occurrences, which are used in the next section of this paper [43].

In conclusion, the introduced techniques have been successfully used in the hydrodynamics analysis in channels with the presence of vegetation patches. However, the reported studies on the hydrodynamics of channels with submerged vegetation patches are limited to those with the flow either above the patches or immediately downstream of the patches. For example, the flow structures was traced up to a distance of $8H_v$ from the toe of the patch towards the downstream [36]. Liu et al. (2018) studied the flow structures to a distance of just 5~6$D_v$ from the patch toe in the downstream direction [28]. The quadrant analysis has been conducted in the field; here, the distance from the patch toe was even more limited, to ~3$D_v$ [38]. Although this distance from the patch toe is enough to address the characteristics of patch-induced turbulence in the near field, it is

not enough to cover the more distant events of turbulence. For example, the investigation of the characteristics of the shear layer generated above the patch requires an extended range of velocimetry in the downstream direction. In addition, there is a need to combine the above-mentioned methods to evaluate the turbulent characteristics of flow behind the submerged vegetation patches. This requires a particular emphasis on the implementation of $\tau_c$ in the turbulence analysis, which has hardly been studied in recent works. In order to fill the above-mentioned gaps in the previous research, in the present study, a combined approach to turbulent analysis is used to investigate the turbulent characteristics of flow in the region downstream of a vegetation patch with an extended range up to $17H_v$.

## 3. Experimental Setup

The experiments were conducted using a glass flume at the Laboratory of Hydraulics of the Iran University of Science and Technology. The flume is 14 m long, 0.9 m wide and 0.6 m deep. The discharge was controlled by an electromagnetic flow meter installed at the entrance of the flume and was set for 31 L/s in the present study. The water level in the flume was adjusted by a tailgate located at the end of the flume and was set for a depth of 18.5 ± 0.3 cm in the present study. The distance between the flume entrance and vegetated flume zone was 6 m to ensure a fully developed flow in the region upstream of the vegetation patch.

The velocity measurements were conducted after the flow reached the steady state condition. The velocity profiles were measured using an acoustic Doppler velocimeter (ADV), placed at the centerline of each row of vegetation patch. There were 19–26 measuring points along each vertical line for velocity measurements, and the vertical distance between two adjacent measuring points was 4–10 mm. The sampling frequency and the measuring time of the ADV were 200 Hz and 120 s, respectively, resulting in 24,000 instantaneous velocities for point measurement.

To date, most previous studies have employed artificial simple rods of regular shapes to simulate natural vegetation patches. As real vegetation is flexible and irregular, this method might not represent the nature of vegetation's behavior [36]. On the other hand, the experiments showed that natural vegetation would lose its stiffness and develop a long-lasting curvature in the flow direction after couple of days. Therefore, a well-shaped synthetic plant was used to represent the natural vegetation, which was selected on the basis of a real world sample of patches in a gravel bed river.

Each model plant had three branches. Each branch had 12 leaves, and the diameter of the branch trunk was approximately 3 mm, as shown in Figure 1. The average height of the vegetation patch was 105 ± 5 mm, and the lateral and longitudinal spread widths of the leaves were approximately 9–19 and 11–22 mm, respectively. The model plants had a certain degree of flexibility and could swing in a flowing current similar to vegetation in a natural river. The vegetation patch was attached to a perforated board in a staggered arrangement. Four different layouts of vegetation patches were used to simulate both fully covered and non-fully covered channel beds. The dimensions of these vegetation patches, namely, length ($L_v$) × width ($D_v$), in this study were 120 × 90 cm, 90 × 60 cm, 60 × 45 cm and 40 × 30 cm, respectively. Table 1 summarizes the experimental runs. The material on the flume bed was a mixture of natural gravel similar to that in a natural gravel bed river (the Marbor River, Zagros Mountains region, Iran). The equivalent particle diameter, which 90% of the total particles were smaller than ($d_{90}$), was 18.8 mm, and $d_{50}$ = 14 mm. The layout of the experimental device and employed materials are shown in Figure 1.

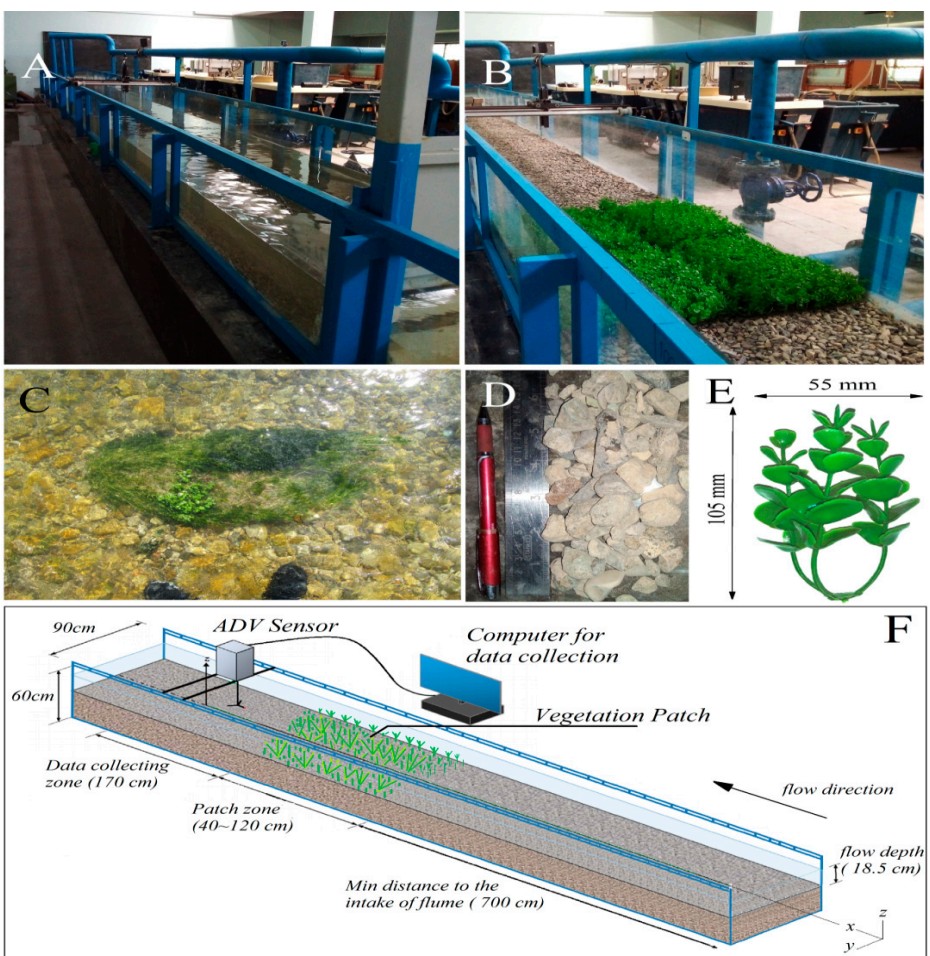

**Figure 1.** (**A**) The layout of the experimental device, (**B**) a sample of vegetation patches (120 × 90 cm, $D_v/D_c$ = 1), (**C**) sample of vegetation patch in a natural gravel bed river (Marbor River, Iran), (**D**) a sample of bed materials, (**E**) a single synthetic plant used to simulate patch; (**F**) a sketch of the experimental setup (not to scale).

**Table 1.** Experimental parameters for the cases discussed in this paper.

| Case | Q (Discharge, L/s) | n/m² (Number of Veg. per Square Meter) | $L_v$ (Length of Patch, cm) | $D_v$ (Width of Patch, cm) | $H_v$ (Height of Patch, cm) |
|---|---|---|---|---|---|
| 1 | 31 L/s | 611.1 | 120 | 90 | 10 |
| 2 | 31 L/s | 611.1 | 90 | 60 | 10 |
| 3 | 31 L/s | 611.1 | 60 | 45 | 10 |
| 4 | 31 L/s | 611.1 | 40 | 30 | 10 |
| No. Veg. | 31 L/s | - | - | - | - |

## 4. Results and Discussion

The results are presented as follows: First, both the velocity profile and the TKE have been investigated to determine the flow layers that formed behind the vegetation patch. Then, spectral analysis and coherent Reynolds shear stress analysis have been carried out to determine the nature of turbulence and coherent Reynolds shear stresses in different flow layers. Consequently, the results of the quadrant analysis of bursting events are provided to compare and evaluate the coherent occurrences in the framework of the time domain. Then, by considering the particular characteristics of turbulence in the mixing layer, the temporal characteristics of turbulence in the mixing layer are investigated. Finally, by

considering the results of previous sections, the transformation of coherent structures in the downstream region of a vegetation patch is provided.

### 4.1. Velocity Profile and TKE

For all experimental runs, three layers of flow can be observed based on the inflection points of the velocity profiles (Figure 2). The associated regions of these layers are known as the wake zone, the mixing layer and the Log–Law shear zone. These three regions resemble the results of trough patch velocimetry reported by other researchers [44–47]. For instance, in case 1 and for a distance from the vegetation edge $x/H_v = 8$, the wake zone formed at the water depth of $z/H_v = 0{\sim}0.7$ ($z/H = 0{\sim}0.56$), the mixing layer formed at the water depth of $z/H_v = 0.7{\sim}1.1$ ($z/H = 0.56{\sim}0.62$) and the Log–Law shear zone formed at the water depth of $z/H_v > 1.1$ ($z/H > 0.62$). The extent and thickness of the mixing layer reduced as the vegetation patch decayed. In this layer, a severe deflection is detectable in the velocity profile, which resembles an adverse pressure gradient. For case 1, this anomalous phenomenon covered a distance (from the vegetation edge) between $8 < x/H_v$ and $x/H_v \geq 17$. In case 2, this phenomenon covered a distance of $3 < x/H_v < 8$, and it was only visible around $x/H_v = 12$ in case 3. However, this phenomenon was absent in case 4. The wake zone was formed in the lower region beyond the patch. For case 1, the wake zone occurred at a distance from the vegetation edge of $x/H_v = 1$, with the upper limit of this layer at $z/H = 0.45$ or $z/H_v = 0.8$, which increased up to $z/H_v = 1$ for the smallest patch (case 4). In this layer, the velocity profiles tended to adopt a Log–Law shape beyond the distance of $x/H_v = 12$ for all cases. For a channel bed partially covered by a vegetation patch, the thickness of the wake layer varied along the flow direction, and increased slightly in such a way that it reached the lower boundary of the Log–Law shear region. Consequently, the mixing layer became narrower and even disappeared in the presence of smaller patches. However, for the case of a channel bed fully covered with a vegetation patch (case 1), the thickness of the mixing layer increased considerably at a distance (from the vegetation edge) of $x/H_v = 8$, and seemed to be effectively present at a distance of $x/H_v \gg 17$.

While the vertical vortexes that normally generate behind a barrier have attracted a lot of attention from researchers, the development of horizontal vortexes (the von Karman Vortex Street) needs to be investigated too. Generally, the formation of von Karman vortexes is associated with the occurrence of peaks in the TKE values [23]. Thus, for each case, the TKE values have been calculated for the relative flow depths of $z/H_v = 0.5$ and $z/H_v = 1$, and also for the near-bed region behind the vegetation patch along the centerline of the flume (Figure 3). The variation in the streamwise velocity along the flow direction is also shown in this figure. In case 1, there was only one peak in the graph of $\text{TKE}/(U_0)^2$ around $x/D_v = 0.9$ and $x/H_v = 7.5{\sim}8$, which is associated with severe deflection in the velocity profile. However, no flow escaped from the sides of the vegetation patch (since the channel bed was fully covered with vegetation). This peak in $\text{TKE}/(U_0)^2$ clearly indicates the presence of a vertical vortex behind the vegetation patch and near the water's surface. While a sharp increase in TKE occured after a distance of $x/H_v = 4.5{\sim}5$, the length of the wake behind the patch can be considered as $L_{vv} = 5H_v$. Importantly, the anomalies in the mixing layer appeared in the same point as was reported [28] for a length of wake of $L_{vv} = (3.5{\sim}5)H_v$ with a wide submerged vegetation patch. However, Folkard (2005) reported a slightly lower range for a wide but highly submerged vegetation patch [29]. Overall, these results are comparable to those of Liu et al. (2018) for a wide and partially vegetation-covered bed with slight submergence [28].

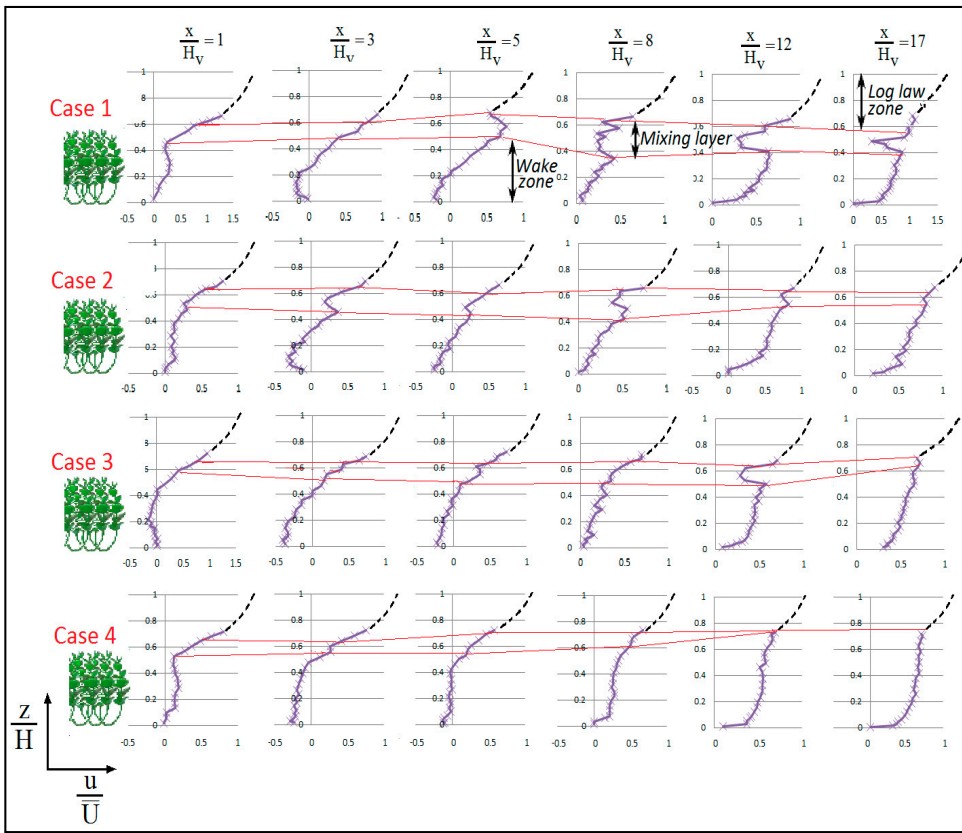

**Figure 2.** Streamwise velocity profiles along the centerline of the flume; $x/H_v$ is the normalized distance from the downstream edge of the vegetation patch, and $H$ is the depth of flow. The boundaries between zones affected by the wake and over the canopy flow are shown as red lines. The velocity of the Log–Law zone is shown as dashed and is not to scale.

For experiment cases 2, 3 and 4, the location of the minimum velocity was shifted towards the flow direction. However, since the patch width ratio of $D_v/D_c$ was relatively high for case 2, $L_{vv}$ was only shifted to the near-bed region. For cases 3 and 4, the shift of wake was also detectable at depths of $z/H_v = 0.5$ and $z/H_v = 1$. In these cases, the shift extent of the upper layers exceeded that of the near-bed zone. For all four cases, the minimum velocity was negative in the near-bed region, implying the occurrence of a vertical vortex. Considering the velocity values in the upper zone, it was observed that the center of the vortex (or rotation) moved upward if $D_v/D_c$ decreased. In case 1, the center of vortex was located in the near-bed region and $z/H_v = 0.5$, while it was located between $z/H_v = 0.5$ and $z/H_v = 1$ in case 4 (smallest vegetation patch). This elevated center of rotation in case 4 pushed the larger part of flow down into the lower zones. Subsequently, the velocity pattern in the middle region became similar to that of the near-bed region (one can observe this by comparing the minimum velocity located between the near-bed region and $z/H_v = 0.5$). In addition, the larger radius of the rotation in the channel with a smaller patch width ratio of $D_v/D_c$ caused the minimum streamwise velocity to be in the near-bed region rather than in the mixing layer. The reason for the upward movement of the rotation center in the case of a smaller patch in the channel may be the higher through-patch velocity. Regarding the larger values of $L_{vv}$ due to the through-patch flow, this statement is consistent with the findings of other researchers [27,28,31]. In addition, for experiment cases 2, 3 and 4, the flow from the sides of the vegetation patches created the second peak in the TKE curve at the depth of $z/H_v = 1$. However, for cases 2 and 3, while the $D_v/D_c$ values were still relatively high, the second peak in the TKE curve did not occur in the middle and near-bed region. Additionally, the location of the single peak in the TKE curve in the lower zones was shifted forward.

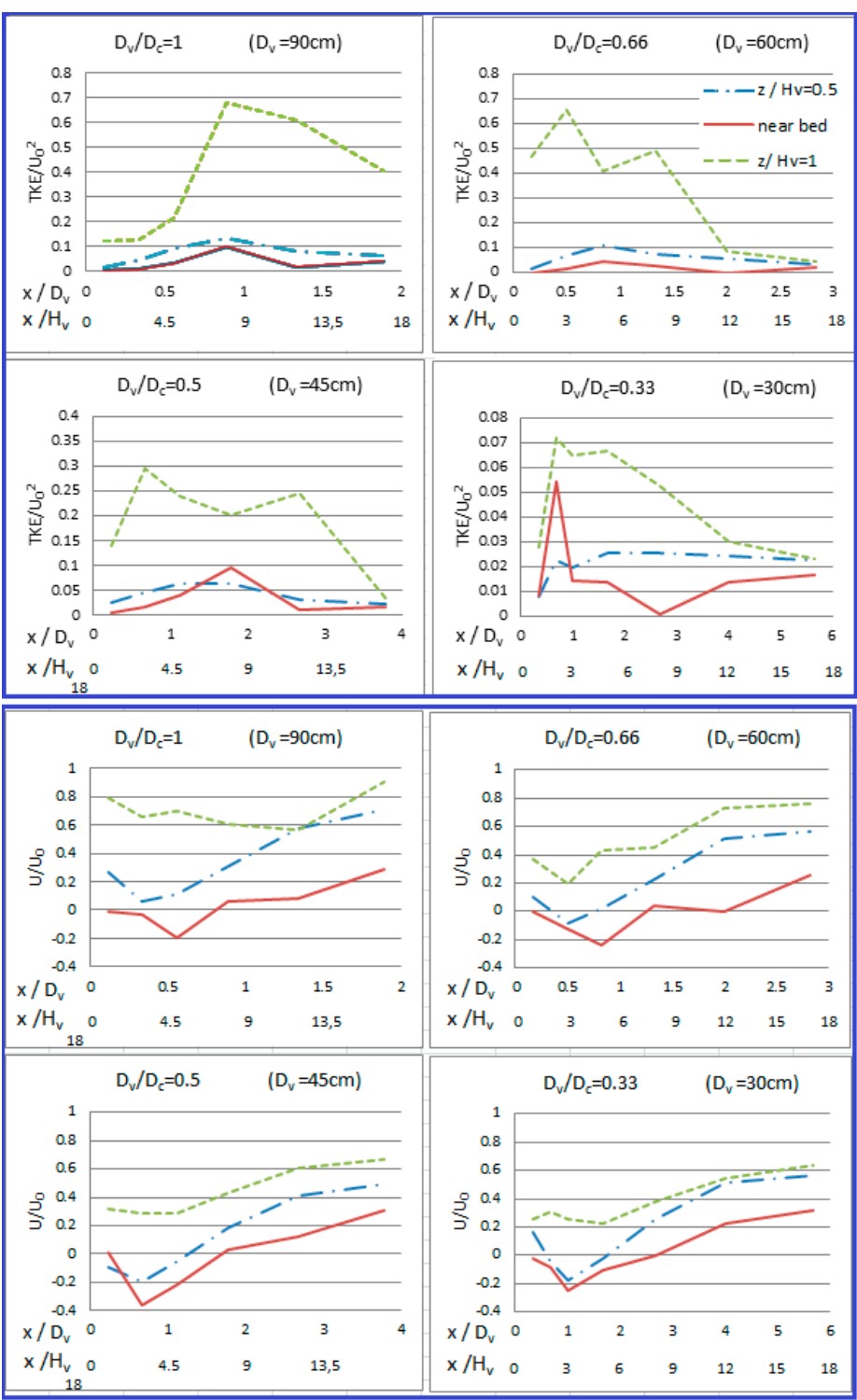

**Figure 3.** Normalized TKE and streamwise velocity behind 4 different patch layouts near channel bed ($z/H_v = 0$), at the middle height of patch ($z/H_v = 0.5$) and top of patch ($z/H_v = 1$).

Overall, the results of both the TKE and velocity analyses show that the three layers (wake, mixing and shear) were present in all cases of vegetation patches. However, the wake and shear layers were considerably affected by the flow passing from the sides of the vegetation patches, and as a consequence, different flow structures and associated length scales emerged, dependent on the upward movement of the center of the vertical rotation beyond the patch.

*4.2. Spectral Analysis and Coherent Reynolds Shear Stress*

The dominant frequency of the streamwise Reynolds shear stress ($f_{d_{u'w'}}$) was the peak of the power spectrum diagram calculated via $u'w'$ fluctuations ($S_{uw}$). Therefore, $S_{uw}$ is calculated at different points for all experimental cases (Figure 4), and the results are used to calculate the coherent Reynolds shear stresses ($-\langle \widetilde{u}_c \widetilde{w}_c \rangle$). The results of the spectral analysis can be placed in the following three categories:

(1)  The first category is the points where the turbulence was well matched with the von Karman isotropic–homogeneous turbulence, but no dominant frequency on $S_{uw}$ was observable. At these points, the power spectra reached the slope of $-5/3$ in the inertial range. As there was no dominant frequency, $f_{d_{u'w'}}$ was too low and no coherent events occurred in the streamwise Reynolds shear stresses (the blank area in Figure 5 represents these points);

(2)  The second category is the points where the turbulence was well matched with the von Karman isotropic–homogeneous turbulence. The dominant frequency on $S_{uw}$ was observable in the form of an obvious peak around 0.07~0.3 Hz. At these points, the power spectra reached the slope of $-5/3$ in the inertial range. Where there was an obvious dominant frequency, $f_{d_{u'w'}}$ was used to calculate the streamwise coherent Reynolds shear stresses. Excluding points at the depth of $z/H_v \approx 1$, all other points for the coherent Reynolds shear stress belong to this group;

(3)  The third category is the points where the turbulence was not compatible with the von Karman isotropic–homogeneous turbulence. The dominant frequency on $S_{uw}$ was observable in the form of an obvious peak around 0.07~2 Hz. At these points, the power spectra reached the slope of $-1$ in the inertial range. According to the analysis provided by Tchen (1953), this slope is associated with the anisotropic turbulence characterized by a large vorticity and strong resonance [47]. These points were observed at a depth of $z/H_v \approx 1$ (mixing layer) for cases 1, 2 and 3, while they were absent in case 4. This finding confirms the upward movement of the rotation center in the presence of smaller patches in the channel (case 4), because the downward velocity beyond the patch was too weak to penetrate into the lower zones and the center of rotation developed in the upper zone. Such a weak vortex cannot produce the strong vertical vortexes required to alter the isotropic turbulence. A Matlab$^@$ code was used to calculate the phase velocity and its deflection for the dominant frequency at each point. The coherent vertical momentum transfer triggered by coherent vertical vortexes continued to appear in both the near-canopy and near-bed regions in cases 1 to 3. However, near-bed coherent shear stress did not occur in case 4. It can be inferred that the flow regime in the near bed region of case 4 was dominated by the flow through the vegetation patch and the wall effect of the rough bed. The presence of the vegetation patch had no coherent effect on this region. In addition, for cases 1, 2 and 3, the majority of coherent occurrences were accompanied by a high spatial gradient of the Reynolds shear stress, which is associated with the boundary layer separation zone and large scale vortexes—a characteristic that was described by Lian (1990) [48]. However, in case 1, the occurrence of coherent Reynolds shear stresses was associated with very low Reynolds shear stresses with a low spatial gradient. Consequently, it can be referred that the coherent Reynolds shear stresses can be classified into two categories according to the origins of the coherence. The first and most prevalent category is the strong coherent Reynolds stress, which was prevalent in the mixing layers of cases 1, 2 and 3. The second category is the weak coherent Reynolds shear stress that occurred in the wake layer. Based on the results of the spectra analysis, the strong coherent shear stresses were associated with peaks of the anisotropic turbulent spectra. In contrast, the weak form of the coherent shear stress occurred in the presence of the peak isotropic turbulence.

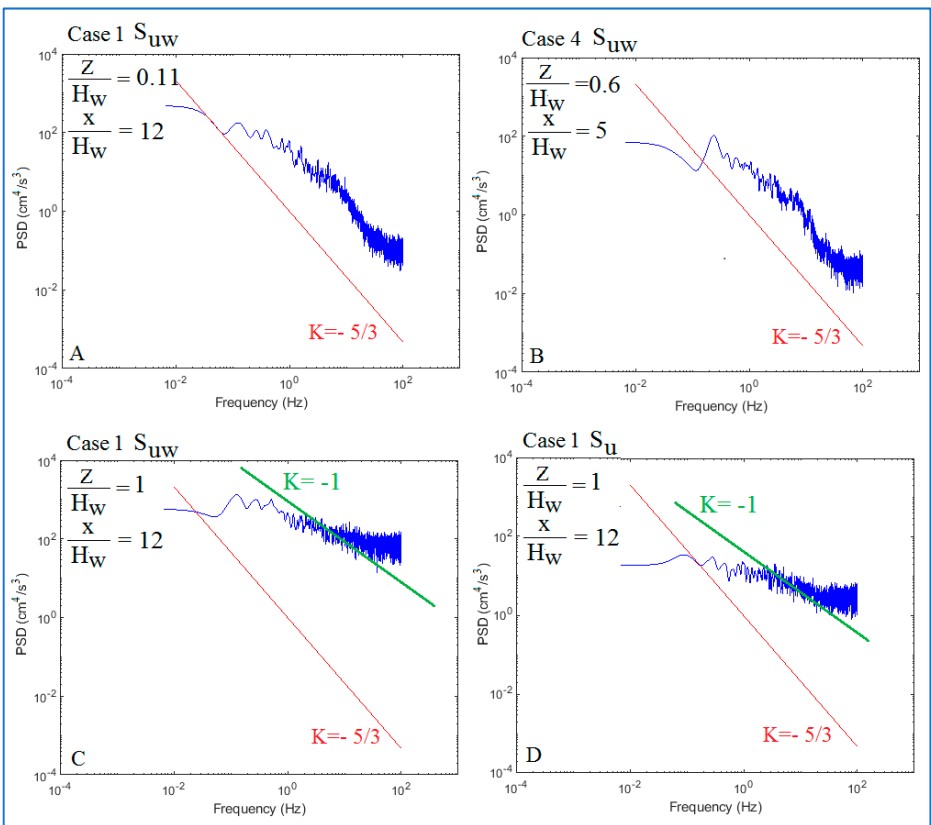

**Figure 4.** Samples of the power spectra behind the patch showing different classes of turbulence: (**A**) isotropic–homogenious turbulent flow without dominant frequency on $S_{uw}$, (**B**) isotropic–homogenious turbulent flow with dominant frequency on $S_{uw}$, observed below the height of the patch, (**C**) anisotropic turbulent flow with dominant frequency on $S_{uw}$, observed alongside the height of patch, (**D**) $S_{uw}$ of the same anisotropic sample.

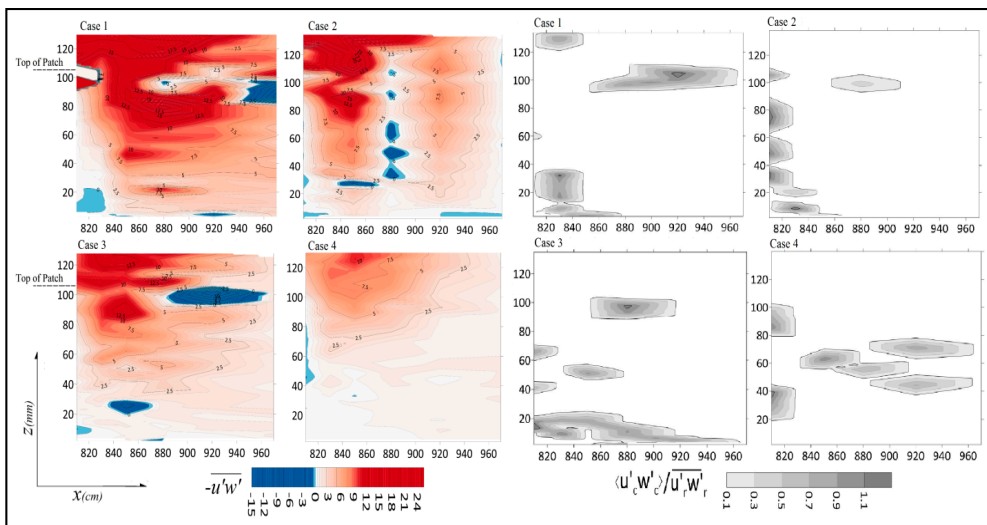

**Figure 5.** Values of $-\overline{u'w'}$ along the centerline of the channel beyond the vegetation patch (colored figures) beside values of $\tau_c/\tau_r$ (graysclae). Note that the blank area in $\tau_c/\tau_r$ is where no dominant frequency was detectable in $S_{uw}$. The end of the patch is located at $x = 800$.

On the other hand, for the patch width ratio of $D_v/D_c = 0.5$, a new type of coherent structure formed in the wake zone, and extended as the patch size reduced. With the patch width ratio of $D_v/D_c = 0.33$ (case 4), the greatest part of wake zone was covered

by this structure. While, in this region, the turbulent spectra of instantaneous velocity resembled isotropic turbulence, the effect of the interaction between the mixing layer and the shear layer seems to be insignificant. On the other hand, the presence of strong flow from the patch sides could form strong von Karman vortexes in the horizontal plane. As reported by Siddique et al. (2008) and Liu et al. (2018), von Karman vortexes appeared in partially covered patches with $H_v/H > 0.55{\sim}0.7$ [28,30]. Consequently, for these regions, the frequency of vortex shedding was calculated as 0.07~0.11 Hz, which is equivalent to an average Strouhal number of about 0.25. These results are comparable with those reported by other researchers [27,28].

### 4.3. Quadrant Analysis of Bursting Events

The occurrence probabilities of the quadrant classes are shown in Figure 6. One can see from Figure 6 that in the presence of smaller patches, the ejection-dominated zone was shifted to a higher elevation at a shorter distance from the patch. This phenomenon was accompanied with lower Reynolds shear stress in the near-bed region. It was also associated with a marginally thicker wake layer in the presence of smaller patches, which was described in the previous section.

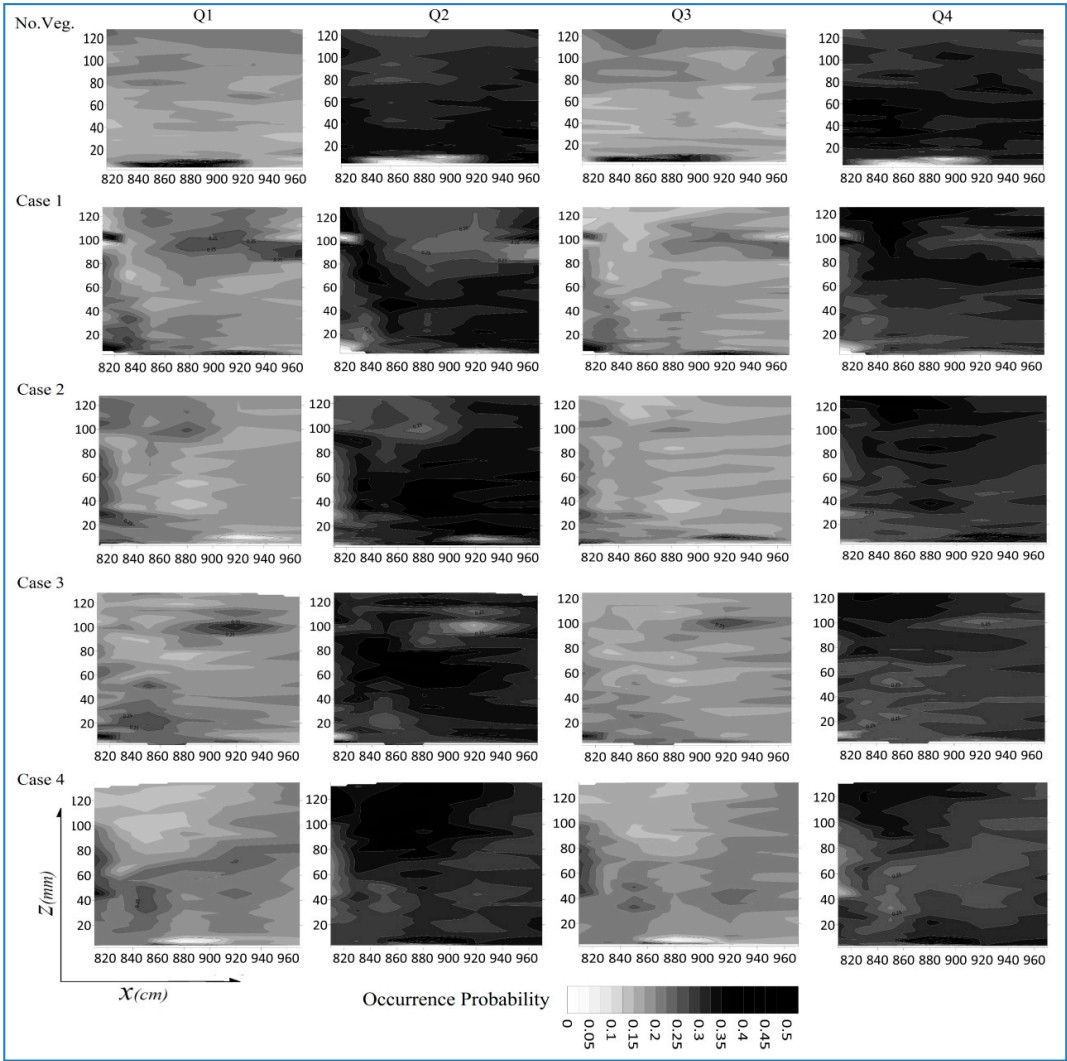

**Figure 6.** Probability of occurrence of quadrant classes behind the patch; the end of the patch is located at $x = 800$. The height of the patches is $105 \pm 5$ mm, and the values are shown up to level of the maximum domain of ADV capability (~140 mm).

In contrast, the sweep-dominated zone, observed above the top of the patch in case 1, faded with the decrease in the patch dimensions. As regards the Reynolds shear stresses of this region, the occurrences of the sweep were associated with the high Reynolds shear stress of the Log–Law layer in the downstream region of the patches. The same pattern was reported in the downstream region of dryland vegetation patches facing wind flow [42].

As strong coherent shear stresses were prevalent in the boundary between the mixing layer and the shear Log–Law layer, the quadrant analysis of the bursting events of this zone are outlined in Figure 7. The propinquity between the outward-dominated regions and regions with coherent Reynolds shear stresses is another notable result of the quadrant analysis. For all vegetation patches, the coherent shear stress in the mixing layer with a frequency of $f_{d_{u'w'}}$ appeared in exactly the same region in which the outward class of quadrant occurrences were dominant, where $f_{k=1} > 0.25$. This was a one-way relationship, and there were some points with $f_{k=1} > 0.25$ at which no coherent shear stress was observable. To assess this important outcome, for point $x$ and quadrant class $i$, two conditional functions can be defined, as below:

$$A(x, i) = \begin{cases} 1 \text{ if } f_{k=i} = \max\{f_{k=1..4}\} \\ 0 \text{ if } f_{k=i} \neq \max\{f_{k=1..4}\} \end{cases} \tag{11}$$

$$B(x) = \begin{cases} 1 \text{ if } f_{d_{(u'w')}} \neq 0 \\ 0 \text{ if } f_{k=i} = 0 \end{cases} \tag{12}$$

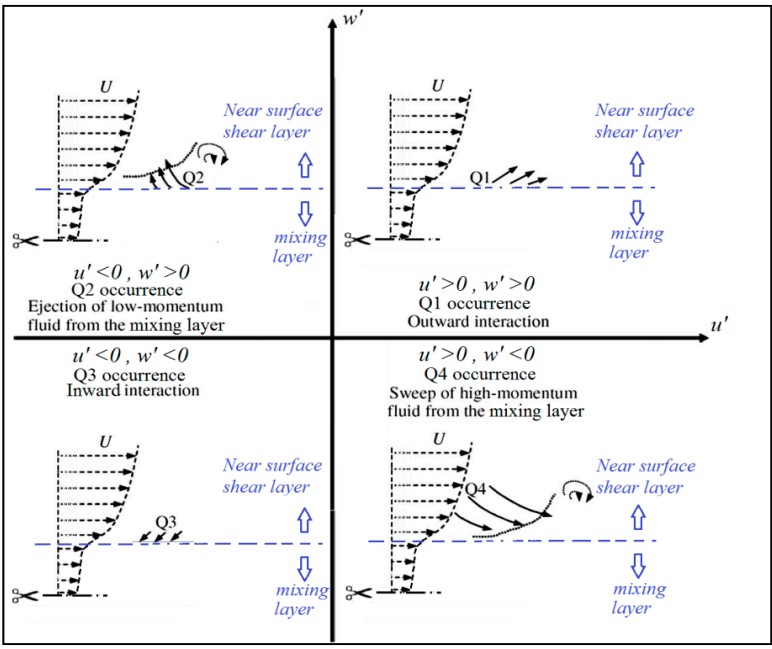

**Figure 7.** Conceptual illustration of the quadratic bursting events in the boundary between the mixing layer and the shear layer.

Subsequently, $\rho_{A,B}$ (the Pearson correlation coefficient between $A(x, i)$ and $(x)$) can be used to determine which class of quadrant occurrences is well-matched with the coherent Reynolds shear stress. For a sample including 278 points for all four cases of patch setup, $\rho_{A,B}$ is 0.68 for Q1 (outward interaction). In comparison, this value is $-0.45$, 0.26 and $-0.38$ for Q2, Q3 and Q4, respectively. This result confirms that the outward interactions are the representative quadrant class of the coherent Reynolds shear stresses in the downstream region of vegetation patches, particularly for strong coherent occurrences. Figure 8 illustrates the values of $f_k$ and $f_{d_{u'w'}}$ for the 278 points beyond the vegetation patches that were investigated in this research. According to the definition of the outward quadrant, the upward momentum flux within the mixing layer toward the shear layer must be considered

(see Figure 7). This upward flux is also detectable in the velocity profiles (Figure 2), where anomalies in the mixing layer are exacerbated; the streamwise velocity was reduced in the mixing layer and increased in the bottom part of the shear layer. This phenomenon illustrates a particular behavior in the velocity time series, which will be discussed in the next section.

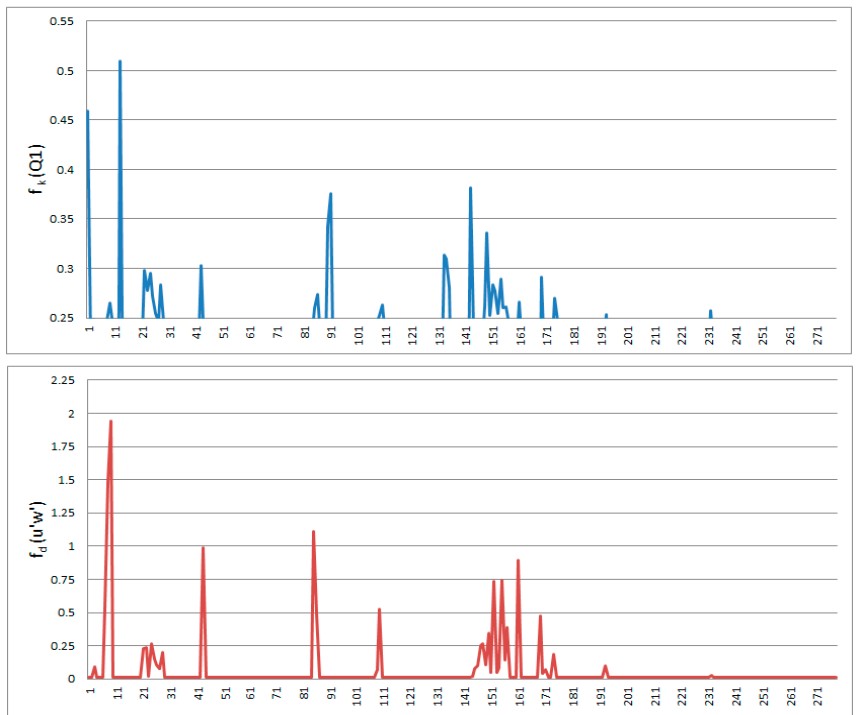

**Figure 8.** Values of $f_k(Q1)$ and $f_{d_{u'w'}}$ for 278 points in the downstream region of the vegetation patches.

### 4.4. Temporal Characteristics of Turbulence in the Mixing Layer

As was described in the previous sections, the mixing layer beyond the vegetation patch is prone to anisotropic turbulence, characterized by outward interaction events and the existence of coherent Reynolds stresses. The deviation from an ideal isotropic turbulence originated from the turbulent intermittency. The turbulent intermittency was characterized by a preference for turbulence with large velocity gradients, which is reflected in the strongly non-Gaussian tails of the probability density functions of velocity differences. These tails are determined by extreme events [49]. Intermittency is also defined as an abnormality in turbulent flow initiated by the interaction between turbulent regions or the interaction between a turbulent region and a vicinal laminar behavior region. The second mechanism is the common source of intermittency in shear flows. Moreover, the occurrence of a highly intermittent region in the near-boundary regions was associated with the prevalence of coherent vortices [50]. The mechanism generates quasi-laminar intervals in the velocity time series of a turbulent flow, which postpones the non-viscous intermediate sub-range in the energy cascade. Thus, there was a slight slope in the energy spectra. Consequently, these intervals persist up to arbitrarily large Reynolds numbers through the energy cascade sequences, which produce extreme events in velocity time series [51]. The determination of a particular intermittency index is a challenging issue and requires the spatial analysis of the velocity time series measured at different locations at the same time. Clearly, there is a need for appropriate sensors, such as PIV and hot wires.

However, temporal analyses of velocity time series can also provide a proper description of the quantity and intensity of intermittent occurrences. As the intermittency is linked with the non-Gaussian tails of the probability density functions of velocity differences, the velocity time series for the mixing layer were subjected to a normality test. The traditional

empirical rule of normality (or a simple normality test) states that, in a normally distributed data set, 68%, 95%, and 99.7% of the values lie within one, two, and three standard deviations ($\sigma$) of the mean ($\mu$), respectively. Most of the points in the wake area could barely pass these restrictions; however, the normality test of Kolmogorov–Smirnov identified almost all points as non-normally distributed data sets. Regarding the methodology and application of the Kolmogorov–Smirnov normality test, please refer to Massey (1951) [52]. The results of the simple and Kolmogorov–Smirnov normality tests for case 1 are shown in Table 2. As the Kolmogorov–Smirnov normality test is very sensitive to non-Gaussian deviations, it even identifies semi-normal velocity time series for the wake layer as non-normal distributions. However, according to the concept of the Kolmogorov–Smirnov normality test, the intensity of non-normality can be observed on the basis of the deviation from the normal cumulative distribution. In this way, an error index can be defined as:

$$Error_{CDF} = 1 - \frac{CDF_E}{CDF_N} \tag{13}$$

where $CDF_E$ refers to the cumulative distribution of the empirical velocity time series, and $CDF_N$ is the cumulative distribution of a normal data set with the same $\mu$ and $\sigma$ as the empirical data set. Figure 9 shows the variation in $Error_{CDF}$ for different depths at a distance of $x/H_v = 8$, where the mixing layer is well developed. One can see that the values of $Error_{CDF}$ are considerably higher at a depth of $0.8 < z/H_v < 1$. However, the most notable issue is that the max values reduce as the vegetation patch disappears. Comparing the results of a simple normality test to $Error_{CDF}$, it can be concluded that a threshold around $Error_{CDF} = 0.2$ can be assumed for the lower limit of intermittency. This threshold is also confirmed by the results of the intermittent turbulence and coherent Reynolds shear stresses at a depth of $0.8 < z/H_v < 1$. Correspondingly, in case 4, where $Error_{CDF} < 0.2$ at all depths, no anisotropy was observed in the turbulent spectra. In contrast, the anisotropic turbulence was limited to the mixing layer in cases 1, 2 and 3. In the smaller $z/H_v$, the no intermittency is observed for all cases and the effect of bed roughness is the determining factor in the near bed region. Generally, in the near bed region of gravel bed channels, the effect of bed roughness exceeds the patch-produced vortexes [53–55].

**Table 2.** Results of the normality test of velocity time series of case 1 at a distance of $x/H_v$=8. Note that the unmatched values are highlighted.

| $z/H_v$ | % of Sample between $\mu - \sigma$ and $\mu + \sigma$ | % of Sample between $\mu - 2\sigma$ and $\mu + 2\sigma$ | % of Sample between $\mu - 3\sigma$ and $\mu + 3\sigma$ | Simple Normality Test | Kolmogorov–Smirnov Test |
|---|---|---|---|---|---|
| Reference Value | 0.68 | 95 | 0997 | | |
| 0.2 | 0.679 | 0.958 | 0.997 | Gaussian | non-Gaussian |
| 0.5 | 0.696 | 0.954 | 0.995 | ~Gaussian | non-Gaussian |
| 0.8 | 0.754 | 0.948 | 0.981 | non-Gaussian | non-Gaussian |
| 1 | 0.710 | 0.938 | 0.998 | non-Gaussian | non-Gaussian |
| 1.2 | 0.753 | 0.928 | 0.993 | non-Gaussian | non-Gaussian |

According to the results of this research, the intermittency and anisotropy of the turbulence are associated with the non-Gaussian distribution of instantaneous velocity. This relationship was also identified by researchers, and is known as a fundamental characteristic of the anisotropic intermittent turbulence [49–51]. However, in the present research, this characteristic was detected in the mixing layer beyond vegetation patches with a relatively high blockage ratio ($D_v/D_c > 0.5$).

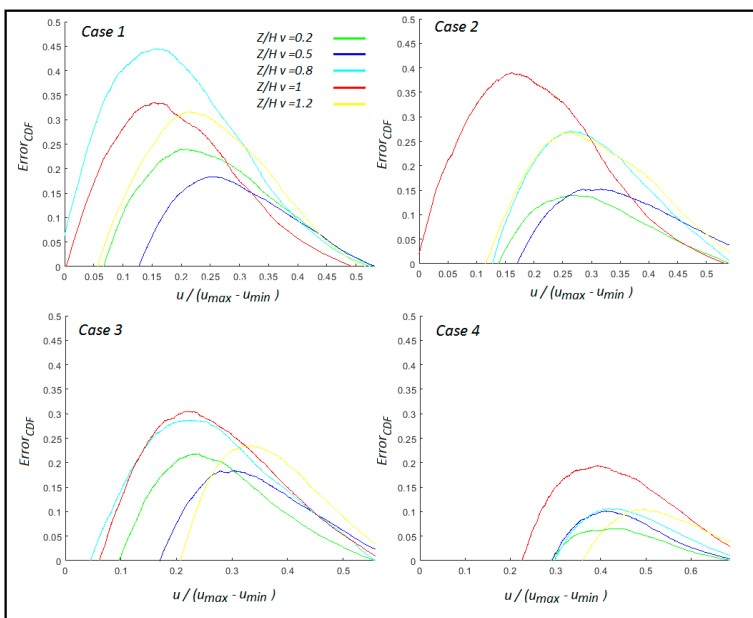

**Figure 9.** Values of $Error_{CDF}$ for different depths behind the decaying patch at a distance of $x/H_v = 8$.

In addition, the formation of intermittent turbulence in the downstream region of the patch has been investigated in this research. One can see from Table 1, by comparing to a normally distributed data set, that the non-Gaussian data sets are dependent on a greater concentration of data between $\mu - \sigma$ and $\mu + \sigma$. In addition, the values of the normalized standard deviation are higher in these velocity time series. Consequently, the variation in the probability density functions (PDF) of the instantaneous velocity was considered along the flow direction. The results have been compared with the turbulent spectra of the corresponding points. For the mixing layer of case 1, these results are shown in Figure 10. As can be seen in Figure 10, the intermittent anisotropic turbulence formed at a depth of $8 < x/H_v < 12$, which is the same range as the occurrence of the outward quadrant dominancy, leading to strong coherent Reynolds shear stresses and anomalies in streamwise velocity. This event can also be inferred from the flattened shape of PDF, which is associated with high standard deviations.

Furthermore, the intensity of the intermittency can also be evaluated in the energy domain. Researchers used an analysis method based on the filtering of TKE time series via a certain threshold [56,57]. These thresholds include $M_{TKE} - a\sigma_{TKE}$ and $M + a\sigma_{TKE}$, where $M_{TKE}$ is the mean, "$a$" is a constant multiplayer (commonly 1 and 3) and $\sigma_{TKE}$ is the standard deviation of the TKE time series in 2D space. The values that exceed this range are considered as strong occurrences characterizing intermittent events. Although the ratio of filtered samples describes the quantity of intermittent events, it cannot provide enough information about the intensity of the intermittency. It is reported that the TKE of extreme events is a suitable indicator of the intermittency [58]. Since extreme events are associated with coherent occurrences in the mixing layer, the Turbulent Kinetic Energy of these extreme events (TKE$_i$) can be used for this propose. This approach is used in this paper for calculating TKE and TKE$_i$ in a 3D space. Overall, Figure 11 summarizes the results regarding the formation of intermittent turbulence in the mixing layer behind the vegetation patch by means of demonstrating the filtered time series of the instantaneous velocity and the variation in the normalized standard deviation of TKE and normalized TKE$_i$.

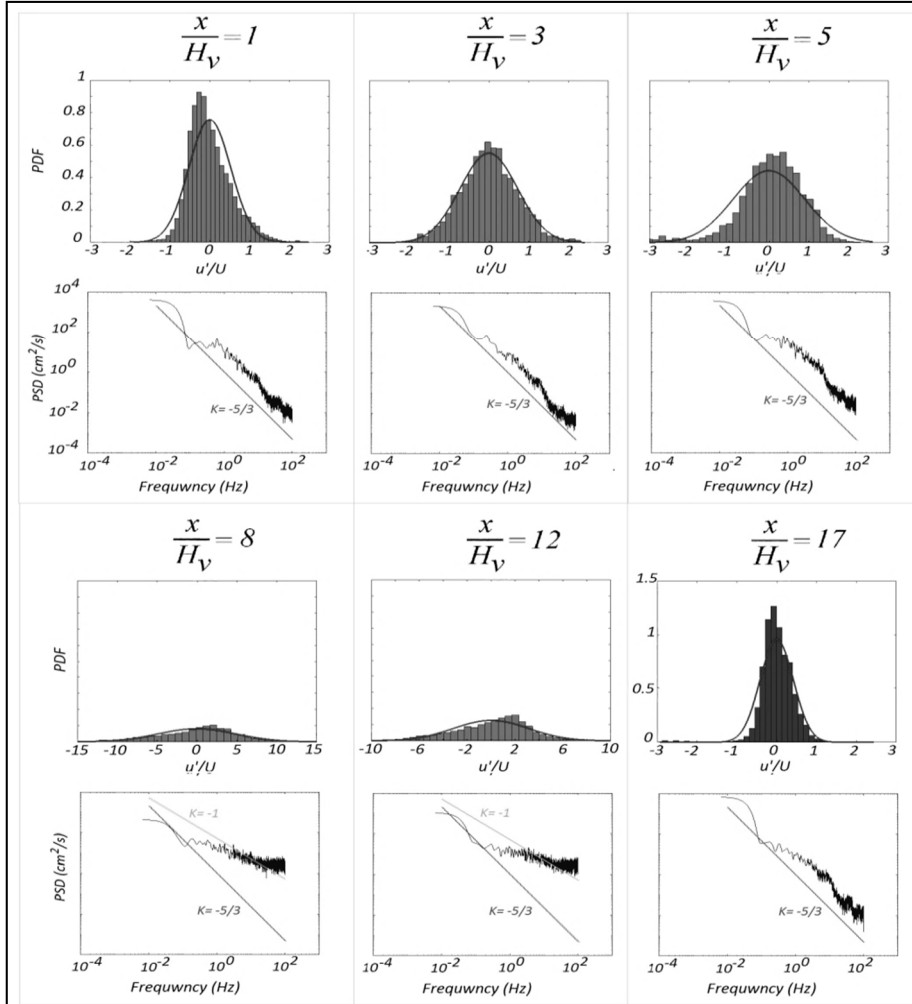

**Figure 10.** Probability density function and the turbulent spectra for the mixing layer of Case 1.

*4.5. Transformation of Coherent Structures beyond a Vegetation Patch*

Considering the results of the previous sections, the transformation of coherent structures beyond a vegetation patch can be summarized as below:

(1) In the downstream region of a fully channel-spanning vegetation patch, the coherent structures are observable just behind the patch. These structures originate from the stem-scale vortexes that are formed by the leaking flow passing through the vegetation patch [24,28]. As the patch width ratio of $D_v/D_c$ reduces to 0.66, the leaking flow increases. Consequently, the stem-scale coherent structures are observed in a larger area (Figure 5). With the patch width ratio of $D_v/D_c = 0.5$, these structures spread to the near-bed region and create a large area of coherent shear stress, in which a Q1-dominant core is surrounded with a sweep ejection-dominant region. However, with a patch width ratio of $D_v/D_c = 0.33$, the area of stem-scale coherent structures reduces suddenly, and is limited to $x/D_v < 3$ in the top and middle zones of the wake layer behind the patch;

(2) In the mixing layer of a fully channel-spanning vegetation patch, coherent shear stresses are associated with largely intermittent fluctuations in the instantaneous velocity forming anisotropic turbulence at $8D_v < x < 12D_v$. In addition, this structure is accompanied by an outward interaction. The intensity and extent of this structure are reduced in the presence of smaller patches, and disappear with a patch width ratio of $D_v/D_c = 0.33$. However, Q1 dominancy is also detectable in the mixing regions of small patches.

(3)   With a patch width ratio of $D_v/D_c = 0.5$, a new type of coherent shear stresses emerges in the wake zone. These structures grow as the dimensions of the vegetation patch reduce, and they come to cover most area of the wake zone with a patch width ratio of $D_v/D_c = 0.33$.

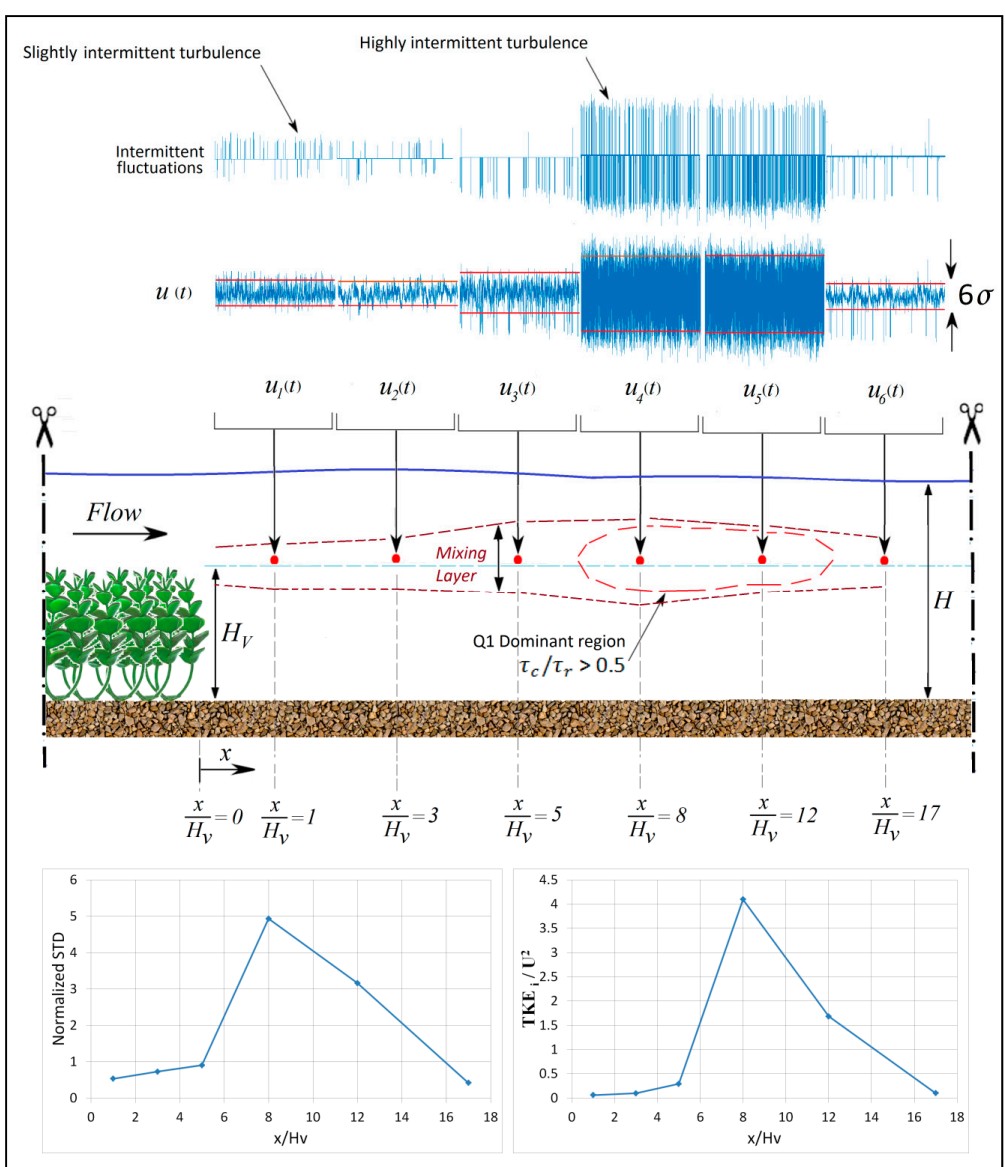

**Figure 11.** Formation of the intermittent fluctuations in the mixing layer beyond the patch (case 1). The filtered velocity time series is shown as intermittent fluctuations above the original velocity time series. Variations in the normalized standard deviation of TKE and normalized intermittent Turbulent Kinetic Energy are shown at the bottom.

In summary, the flow in the downstream region of a decaying vegetation patch alters as the size of vegetation patch decreases. This change includes not only the intensity of occurrences, but also the nature of structure. This means that the dominant coherent structures are transferred from the intermittent anisotropic turbulent fluctuations in the mixing layer of a fully covered patch into the isotropic turbulence associated with the von Karman vortexes on the horizontal plane of the wake layer in the presence of the smallest patch.

## 5. Conclusions

In the present study, the changes in turbulent structures in the regions downstream of vegetation patches have been investigated. The model vegetation was selected carefully to simulate aquatic vegetation patches in natural rivers. Velocity profile, TKE, turbulent power spectra and quadrant analysis have been used via various approaches to investigate the features and intensity of the turbulent structures. Three different flow layers were detected in the downstream regions of vegetation patches, including the wake layer, the mixing layer and the shear layer. Overall, the results of the TKE and velocity analysis show that those three flow layers (wake, mixing and shear) occurred in all cases of patch setup. However, the wake and shear layers were considerably affected by the flow passing around the sides of the patches. Consequently, different flow structures and associated length scales can be formed, which are dependent on the upward movement of the center of the vertical rotation beyond the patch. The results of spectral analysis indicate that the strong coherent shear stresses are associated with the peaks of the anisotropic turbulent spectra. In contrast, the weak form of coherent shear stresses occurs in the presence of peak isotropic turbulence. The presence of von Karman vortexes near partially covered patches at a depth of $H_v/H > 0.55\sim0.7$ shows that the frequency of vortex shedding was around 0.07–0.11 Hz, which is equivalent to an average Strouhal number of about 0.25. The main characteristics of turbulent structures beyond a wide patch are associated with the highly intermittent anisotropic turbulent events in the mixing layer, and appear at a depth of $z/H_v = 0.7\sim1.1$ and distance of $x/H_v = 8\sim12$. The outward interactions are the representative quadrant class of coherent Reynolds shear stresses in the regions downstream of vegetation patches—particularly the strong coherent occurrences. The intensity and extent of these structures decrease as the size of the patch reduces. In addition, the intermittency and anisotropy of the turbulence are associated with the non-Gaussian distribution of the instantaneous velocity detected in the mixing layer beyond patches with a relatively high patch width ratio of $D_v/D_c > 0.5$. Finally, when the size of the patch reduces, von Karman vortexes appear in the wake layer, and form the dominant flow structures in the downstream region of the vegetation patch. It should be noted that the results of this research are limited by our use of a certain type of vegetation, and so it is strongly suggested that other types of the vegetation patches often found in natural rivers be considered in future research.

**Author Contributions:** M.K.; laboratory works, methodology, software and writing—original draft preparation, H.A.; laboratory supervisory, methodology and validation, J.S.; methodology and writing reviewing. All authors have read and agreed to the published version of the manuscript.

**Funding:** This research received no external funding.

**Institutional Review Board Statement:** This study did not involve humans or animals.

**Informed Consent Statement:** Not applicable.

**Data Availability Statement:** The data are available upon request.

**Acknowledgments:** Special thanks to G. Goodrati Amiri for his contribution in providing the laboratory devices and Esmaeel Dodangeh for his contribution in laboratory setup.

**Conflicts of Interest:** The authors declare no conflict of interest.

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
