# Peer review of "Characteristics of Turbulence in the Downstream Region of a Vegetation Patch"

_water, doi:10.3390/w13233468_

Round 1

Reviewer 1 Report

In this article, the authors extensively studied the impact of vegetation patches on the flow patterns. The experimental procedure and data investigation appear to be detailed and thorough. The conclusion was also nicely presented and seems to be convincing. However, there are quite some editing problems, which indicates the paper needs careful proofreading and correction before acceptance. Detailed comments are given below:

  1. It is suggested to include a short introduction and the general impact of the research in the abstract.
  2. The meaning of “TKE” in the abstract needs to be explained upon the first usage.
  3. Symbol inconsistency. On line 116, the symbol “u’” is not used in Eq. (4).
  4. Symbol inconsistency. On lines 132 and 134, it seems the authors are using the letter “t” to denote the Greek symbol “tau” in Eqs. (7-8), which is quite confusing. This also seems true in later discussion, such as line 198.
  5. In Eq. (9), the meaning of “nk” should be explained, although it might be straightforward.
  6. Symbol inconsistency. On line 153, the authors used “CH” to denote the threshold parameter, which is not the format used in Eq. (10).
  7. Figure 3 – 4, the texts are stretched significantly. Some of the text fonts are too small to be seen.

Author Response

We would like to thank the reviewer for the insightful comments and constructive suggestions, which help us to improve the quality of this manuscript. Our response follows:

Comment 1: It is suggested to include a short introduction and the general impact of the research in the abstract

--Response: Thank you for your comment. Following sentences have been added to the abstract part:  “In presence of vegetation patches in channel bed, different flow-morphology interactions in a river will be resulted. The investigation on the nature and intensity of these structures is a crucial part of the research works of river engineering.”

Comment 2: The meaning of “TKE” in the abstract needs to be explained upon the first usage.

--Response: Thank you for pointing this out. “Turbulent Kinetic Energy” is added in the parentheses as an explanation just after the TKE term in the abstract part.

Comment 3: Symbol inconsistency. On line 116, the symbol “u’” is not used in Eq. (4).
--Response: In the revised version, symbol “ u(x,t) “ is defined just after Eq(4).

Comment 4: Symbol inconsistency. On lines 132 and 134, it seems the authors are using the letter “t” to denote the Greek symbol “tau” in Eqs. (7-8), which is quite confusing. This also seems true in later discussion, such as line 198.

--Response: Thank you for pointing this out. It has happened because of format transfer. Sorry for that. All symbols are checked and correct in the revised version.

Comment 5: In Eq. (9), the meaning of “nk” should be explained, although it might be straightforward.
--Response: An explanation is added in the revised version: “Where, n_k is the number of events belonging to class k, and N is the total number of events.”

Comment 6: Symbol inconsistency. On line 153, the authors used “CH” to denote the threshold parameter, which is not the format used in Eq. (10).
--Response: Thank you for pointing this out. It is corrected in the revised version.

Comment 7: Figure 3 – 4, the texts are stretched significantly. Some of the text fonts are too small to be seen.
--Response: Thank you for pointing this out. Both Figures3 and 4 are refined in the revised version.

Reviewer 2 Report

Investigation of turbulent structure changes in the downstream region of vegetation patches was presented in this paper. Velocity profile, TKE, turbulent power spectra, and quadrant analysis was employed to investigate the features and intensity of the turbulent structures. Experiment results are also provided.  In general, this paper is well written and the topic is interesting. Here, there are some concerns of this reviewer:

Comments to the Authors:

  1. Index Terms/ Keywords:

(a). The keywords should be arranged in alphabetic order.

  1. Introduction/ Related works:

(a). Introduction can be strengthened. Please note that the up-to-date references will contribute to the up-to-date of your manuscript.

  1. Experimental setup:

Line 239: Author(s) mentioned figure 1 but the figure label read Figure 1. Please double check

  1. Results and discussion

Line 528: Author(s) stated that Figure 11 summarized the results about the ……… I can’t find Figure 11 in the manuscript. Please double-check your figures. It seems Figure 10 appeared twice.

  1. Conclusions

The authors have not presented the limitations of this work in practical applications. How can this work be extended in the future?

Author Response

We would like to thank the reviewer for the insightful comments and constructive suggestions, which help us to improve the quality of this manuscript. Our response follows:

Your comment: Index Terms/ Keywords (Comment 1):  The keywords should be arranged in alphabetic order.

---Response: Thank you for pointing this out. It is corrected in the revised version.

Your comment: Introduction/ Related works (Comment 2): Introduction can be strengthened. Please note that the up-to-date references will contribute to the up-to-date of your manuscript.

---Response: A number of the newest references are added in the introduction of the revised version. However, a considerable number of the current papers are cited in the “Results and discussion” section.

Your comment: Experimental setup (Comment 3): Line 239: Author(s) mentioned figure 1 but the figure label read Figure 1. Please double check

---Response: Thank you for pointing this out. It is corrected in the revised version.

Your comment: Results and discussion (Comment 4): Line 528: Author(s) stated that Figure 11 summarized the results about the ……… I can’t find Figure 11 in the manuscript. Please double-check your figures. It seems Figure 10 appeared twice.

---Response: Thank you for pointing this out. Figure 11 was wrongly labeled as Figure 10 (there were two Figure 10).  It is corrected in the revised version.

Your comment: Conclusions (Comment 5): The authors have not presented the limitations of this work in practical applications. How can this work be extended in the future?

---Response: In the conclusion section of the revised version, the following sentences are added: “It should be noted that results of this research are limited by using a certain type of vegetation, it is strongly suggested that other types of vegetation patches often found in natural rivers must be considered in the future researches.”